# Hybrid Nanowire–Rectangular Plasmonic Waveguide for Subwavelength Confinement at 1550 Nm

**DOI:** 10.3390/mi13071009

**Published:** 2022-06-26

**Authors:** Yindi Wang, Hongxia Liu, Shulong Wang, Ming Cai

**Affiliations:** Key Laboratory for Wide Band Gap Semiconductor Materials and Devices of Education Ministry, School of Microelectronics, Xidian University, Xi’an 710071, China; wangyindi4213@126.com (Y.W.); cm9999787@163.com (M.C.)

**Keywords:** waveguide, SPPs, subwavelength

## Abstract

This paper presents a hybrid waveguide based on metal surface plasmon polaritons (SPPs) at 1550 nm comprising two silver (Ag) nanowires and a rectangular silicon (Si) waveguide. Due to the strong coupling effect observed in both the metal SPP mode and Si waveguide mode, excellent waveguide characteristics, such as a small effective modal area and long transmission length, could be achieved. The research results revealed that the proposed hybrid waveguide could achieve an ultra-long transmission distance of 270 µm and normalized effective mode area of 0.01. Furthermore, the cross-sectional size of the waveguide was 500 nm × 500 nm, which helped in achieving a subwavelength size. In addition, the hybrid waveguide was resistant to manufacturing errors. These excellent performances indicate that the proposed waveguide has great application potential in optoelectronic integrated circuits.

## 1. Introduction

With the recent advances in the field of optical communications, there has been an increase in the performance requirements of waveguides [1,2,3,4,5]. In particular, there have been rapid advances in optoelectronic integrated systems. These systems require smaller optical devices [6] to improve the system integration. Therefore, the waveguides require both a long transmission distance and small normalized mode area [7,8,9]. In this context, surface plasmon polaritons (SPPs) can be introduced into the waveguide design, which will help in improving the normalized effective mode area of the waveguide [10,11,12], thereby reducing the size of waveguide-based optical devices.

SPPs are caused by the resonance excitation of incident photons with free electron gas on a metal surface [13,14,15,16]. They are widely used for optical manipulation at the subwavelength scale because they can restrict light to a considerably smaller range than the diffraction limit [17,18]. Currently, the most widely used precious metal SPPs are gold (Au) and silver (Ag) [19]. In recent years, several types of waveguides have been designed based on SPPs [20,21,22], and the normalized effective mode area of waveguides has been significantly improved [10,24,25]. However, further research is required to simultaneously ensure a long transmission distance and small normalized effective mode area.

In this study, we developed a long-range hybrid waveguide operating at 1550 nm. The hybrid waveguide is composed of two Ag nanowires and a silicon (Si) waveguide. The waveguide structure has two coupling regions, which strengthens the mode coupling. Due to the strong coupling between the Ag SPPs and Si medium, the hybrid waveguide exhibits both a compact mode area and an extremely long transmission distance. The cross-sectional area of the waveguide is 500 nm × 500 nm, and a subwavelength structure is realized. Moreover, the proposed hybrid waveguide is resistant to manufacturing errors and has a wide range of applications in optical communication systems owing to its advantages.

## 2. Structure Design

The hybrid waveguide consists of two Ag nanowires and a conventional Si waveguide structure. The entire structure is filled with SiO_2_ cladding, which acts as a low exponential gap between Si and Ag [23]. Excellent waveguide characteristics were obtained as a result of the double-mode coupling of Ag and Si in this structure.

The 3D structure and cross-section of the proposed hybrid waveguide are shown in Figure 1a,b, respectively. The cross-sectional area of the waveguide was 500 nm × 500 nm. Note that *r_Ag_*, *T_Si_*, and g denote the radius of the Ag nanowire, thickness of the rectangular Si waveguide, and gap height between the Ag nanowire and Si waveguide, respectively.

## 3. Methods

The proposed hybrid waveguide utilizes the strong coupling of the Si waveguide and SPP modes to achieve good waveguide performance. It supports an ultra-long transmission distance and has an ultra-compact effective mode area. More importantly, there are two couplings in the hybrid waveguide structure. The superposition of these two couplings resulted in stronger light field localization, which helped improve the waveguide characteristics.

The properties of the hybrid waveguide can be characterized by the following four parameters: *A_eff_*, *L_m_*, *FOM*, and *n_eff_* [19,24].

Here, *A_eff_* is the effective modal area; its value determines the optical field constraint capability of the waveguide [25]. A smaller value of *A_eff_* indicates that the light is more localized in the waveguide [26]. *A_eff_* is the ratio of the total electromagnetic energy to the maximum energy density [18]. It is calculated as follows:(1)$$A_{eff}=\iint W\left(r\right)dA/\max\left(W\left(r\right)\right)$$
(2)W(r)=12Re{d[ωε(r)]dω}|E(r)|2+12μ0|H(r)|2
where *H(r)* and *E(r)* represent the magnetic and electric fields, respectively. In this study, the normalized effective mode area was used; it is calculated using the formula *A_eff_*/*A*_0_, where *A*_0_ = *λ*^2^/4 [27].

Note that L_m_ represents the transmission length of the hybrid waveguide, and it is defined as follows [28]:(3)Lm=12α=λ/4πIm(Neff)
where *N_eff_* denotes the effective modal index of the waveguide. Note that *A_eff_* and *L_m_* represent two mutually restrictive parameters, and a smaller effective modal area is often accompanied by a shorter transmission length. *FOM* is the quality factor of the waveguide, which is a tradeoff between *A_eff_* and *L_m_*. *FOM* is calculated using the following equation:(4)$$FOM=\frac{L_{m}}{2\sqrt{\frac{A_{eff}}{\pi }}}$$
where *N_eff_* represents the loss of light in the waveguide, and *n_eff_* represents the real part of *N_eff_*, which can be defined as follows:*n_eff_* = *Re*(*N_eff_*)(5)

## 4. Results and Discussion

The hybrid waveguide utilizes the mode coupling of the Si photon and Ag nanowire plasma modes, thereby achieving excellent waveguide characteristics. Strong mode coupling can be obtained easily by adjusting the key structural parameters of the hybrid waveguide, including *g*, *r_Ag_*, and *T_Si_*.

COMSOL Multiphysics software was used for the numerical analysis of the parameters described in the methods section. In the following simulations, the incident wavelength was 1550 nm, and the relative dielectric constants of Ag, Si, and SiO_2_ were −129 + 3.3i, 12.25, and 2.25 [18], respectively. In this study, the classical control variable method was adopted to obtain the relationship between the waveguide characteristics and parameters to optimize the waveguide characteristics.

First, the relationship between the waveguide characteristics and *g* was studied. Herein, *g* had a range of 1–20 nm. To ensure effective mode coupling and moderate transmission loss, *r_Ag_* and *T_Si_* were fixed at 30 and 20 nm, respectively.

As illustrated in Figure 2, with an increase in g, the normalized effective mode area, transmission length, and *n_eff_* increased, whereas FOM decreased. The main reason for these results is that the coupling of the hybrid mode weakened with increasing g. The weakened mode coupling resulted in reduced light localization and an increase in the effective mode area. In turn, the length of the light transmission increased. The proposed hybrid waveguide had an extremely long transmission length (*L_m_*: 180–270 µm) while maintaining a very small normalized effective mode area (*A_eff_*/*A*_0_: 0.01–0.05). *FOM* is a compromise between the effective mode area and transmission length. From its calculation formula, it can be observed that the *FOM* and transmission length exhibit the same change trend, and the proposed hybrid waveguide has a very high-quality factor (*FOM*: 1350–2000).

To further verify the abovementioned calculation results, the electric field distributions of the waveguide at different gap heights (i.e., different values of *g*) are presented. As shown in Figure 3, with an increase in *g*, the electric field distribution was increasingly diffused. The electric field was most concentrated at *g* = 5 nm, and it was concentrated in the gap region. As g increased, the electric field gradually spread beyond the gap region. This result is consistent with that shown in Figure 2. In this study, considering the characteristics of the waveguide and the actual manufacturing process, a value of 10 nm was selected for *g* in the following simulations.

Subsequently, the influence of different Si thicknesses (i.e., different values of *T_Si_*) on the waveguide characteristics was investigated. In this calculation, the value of *T_Si_* varied in the range of 1–30 nm, *r_Ag_* was 30 nm, and *g* was 10 nm. The results are shown in Figure 4, wherein, it can be observed that the characteristics of the hybrid waveguide were only slightly affected by the Si thickness because it had no effect on the coupling of the two hybrid modes.

To verify the above conclusions, the electric field distributions for different Si thicknesses were obtained. Here, Figure 5a–c show the electric field distributions at different values of *T_Si_*. It can be observed that for different Si thicknesses, the electric field of the waveguide did not diffuse, and the light was well localized. This is consistent with the calculation results shown in Figure 4. Therefore, we fixed *T_Si_* at 20 nm in the following simulations.

Subsequently, the effect of the radius of the Ag nanowires (*r_Ag_*) on the waveguide characteristics was explored. Here, *r_Ag_* had a range of 10–100 nm, and g and *T_Si_* were fixed at 10 and 20 nm, respectively. Figure 6a–d show the dependence of *A_eff_*/*A*_0_, *L_m_*, *FOM*, and *n_eff_* on *r_Ag_*, respectively. It can be observed that the effective mode area (*A_eff_*/*A*_0_) was very large, and the transmission distance (*L_m_*) was short in the region where *r_Ag_* was less than 20 nm and *r_Ag_* was greater than 75 nm. The primary reason for this phenomenon is that no mode coupling occurred in this region. Furthermore, the waveguide exhibited good characteristics when *r_Ag_* was in the range of 20–75 nm. This is due to the strong mode coupling occurring in this region.

Here, Figure 6e–g show the electric field distributions at *r_Ag_* = 15, 30, and 80 nm, respectively. It can be observed that at *r_Ag_* = 15 and 80 nm, no coupling occurred, and the electric field was dispersed in the cladding, whereas at *r_Ag_* = 30 nm, mode coupling occurred, and the electric field was concentrated in the gap region. This is consistent with the calculation results shown in Figure 6a–d.

Figure 7 shows the electric field distributions at different values of *r_Ag_* (20–70 nm). As *r_Ag_* increased, the electric field became more localized because a larger radius of the Ag nanowires leads to stronger mode coupling. Considering this observation in combination with the results observed in Figure 6, a value of 30 nm was selected for *r_Ag_* in the following simulations.

Based on the aforementioned results, it can be concluded that the characteristics of the waveguide can be optimized by adjusting the geometric parameters. Thus, the proposed hybrid waveguide exhibited a small effective mode area and an ultra-long transmission distance.

## 5. Fabrication Processing and Fabrication Error Tolerance

After all the geometric parameters were determined, a hybrid waveguide was fabricated using the standard micromachining process [29,30]. All the geometric parameters of the waveguide could be precisely controlled, except for the central symmetry of the two Ag nanowires. Next, we investigated the difference in waveguide characteristics caused by the deviation of the center position of the two Ag nanowires (*dx*). In this simulation, *g*, *r_Ag_*, and *T_Si_* were 10, 30, and 20 nm, respectively. As shown in Figure 8, the center asymmetry error, *dx* (1–30 nm), caused errors of only 7% and 2.4% for *A_eff_*/*A*_0_ and *L_m_*, respectively.

Figure 9 shows the electric field distributions at different values of *dx* (10, 20, and 30 nm). As *dx* increased, the waveguide maintained strong mode coupling and good electric field localization, which is consistent with the results observed in Figure 8. These results showed that the hybrid waveguide is reliable and robust against manufacturing errors.

## 6. Conclusions

In this study, a hybrid waveguide based on metal SPPs and operating at 1550 nm was proposed. The hybrid waveguide achieved both a small effective modal area and long transmission length using the strong coupling between the metal SPP mode and Si waveguide mode. The results showed that the waveguide achieved an ultra-long transmission distance of 270 µm and effective mode area of 0.01. The cross-sectional size of the waveguide was only 500 nm × 500 nm, which realized a subwavelength size. The results showed that the properties of the proposed waveguide are resistant to manufacturing errors. These excellent performances indicate that the proposed waveguide has great application potential in optoelectronic integrated circuits.

## Figures and Tables

**Figure 1 micromachines-13-01009-f001:**
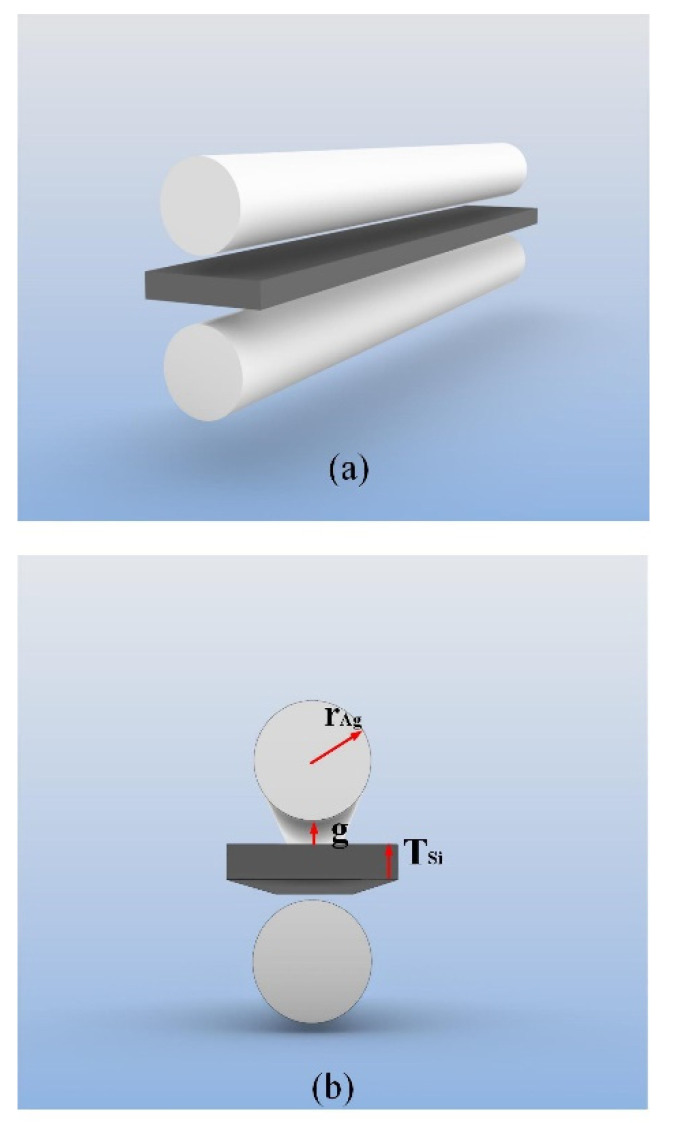
Structure of the proposed waveguide: (**a**) 3D layout and (**b**) cross-section.

**Figure 2 micromachines-13-01009-f002:**
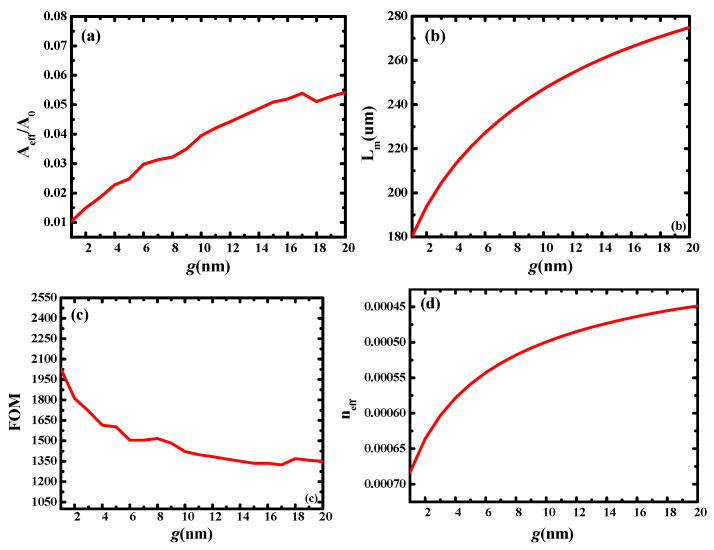
Dependence of waveguide characteristics on gap height (*g*): (**a**) normalized mode area (*A_eff_*/*A*_0_); (**b**) propagation length (*L_m_*); (**c**) quality factor of waveguide (*FOM*); and (**d**) modal effective index (*n_eff_*).

**Figure 3 micromachines-13-01009-f003:**
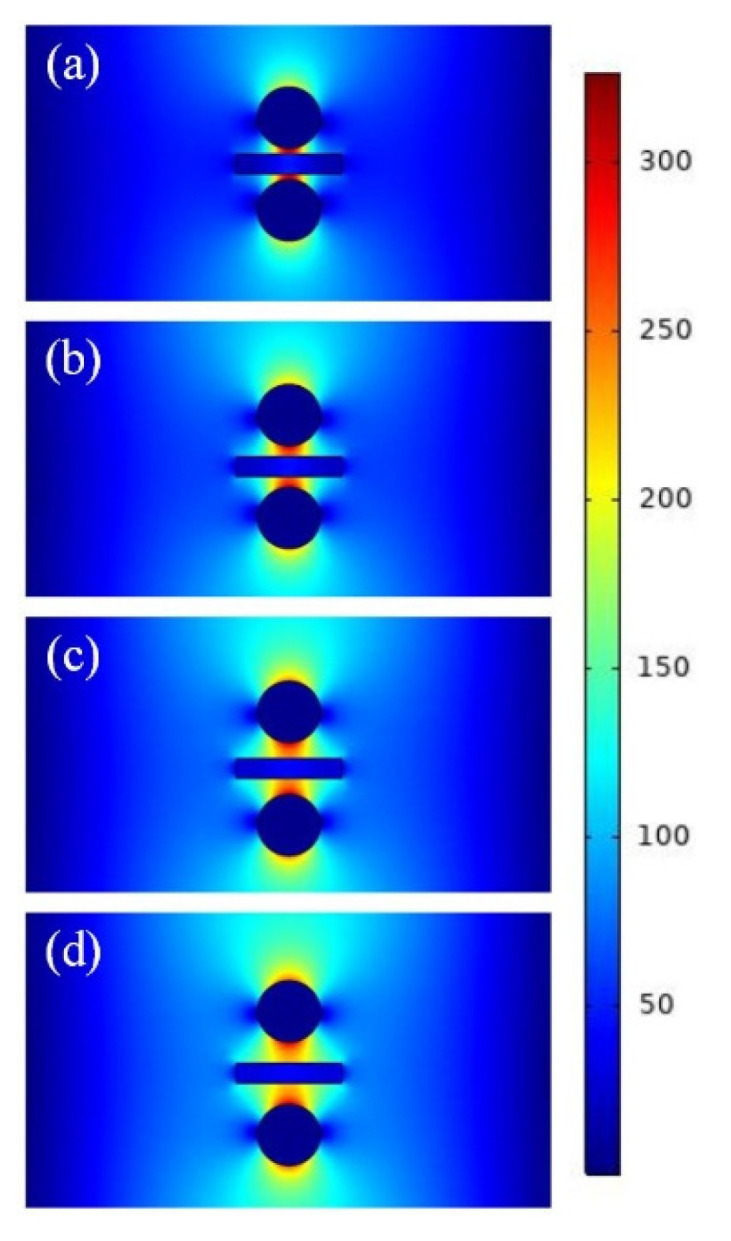
Electric field distributions of waveguide at different gap heights (*g*): (**a**) *g* = 5 nm; (**b**) *g* = 10 nm; (**c**) *g* = 15 nm; and (**d**) *g* = 20 nm.

**Figure 4 micromachines-13-01009-f004:**
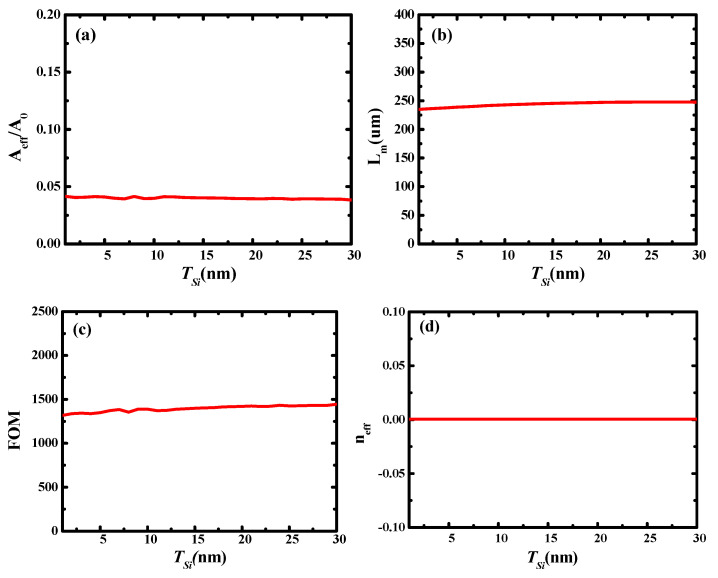
Dependence of waveguide characteristics on Si thickness (*T_Si_*): (**a**) normalized mode area (*A_eff_/A*_0_); (**b**) propagation length (*L_m_*); (**c**) quality factor of waveguide (*FOM*); and (**d**) modal effective index (*n_eff_*).

**Figure 5 micromachines-13-01009-f005:**
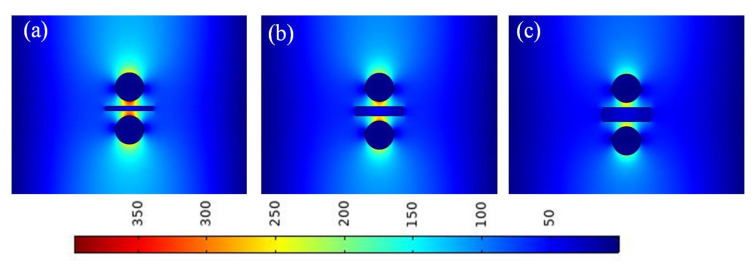
Electric field distributions at different Si thicknesses (*T_Si_*): (**a**) *T_Si_* = 10 nm; (**b**) *T_Si_* = 20 nm; and (**c**) *T_Si_* = 30 nm.

**Figure 6 micromachines-13-01009-f006:**
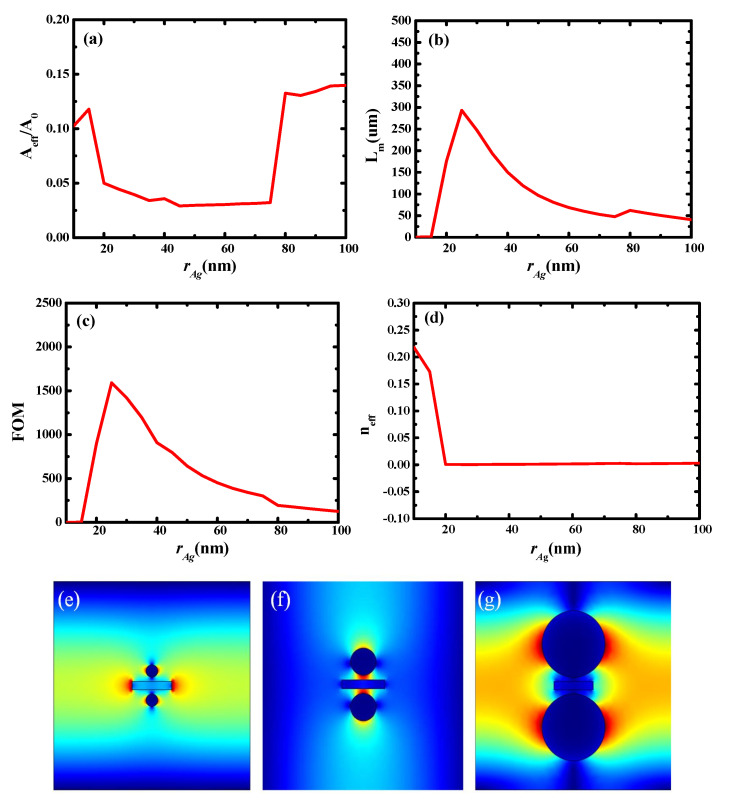
Dependence of waveguide characteristics on radius of Ag nanowires (*r_Ag_*): (**a**) normalized mode area (*A_eff_*/*A*_0_); (**b**) propagation length (*L_m_*); (**c**) quality factor of waveguide (*FOM*); (**d**) modal effective index (*n_eff_*); (**e**) electric field distributions at *r_Ag_* = 15 nm; (**f**) electric field distributions at *r_Ag_* = 30 nm; and (**g**) electric field distributions at *r_Ag_* = 80 nm.

**Figure 7 micromachines-13-01009-f007:**
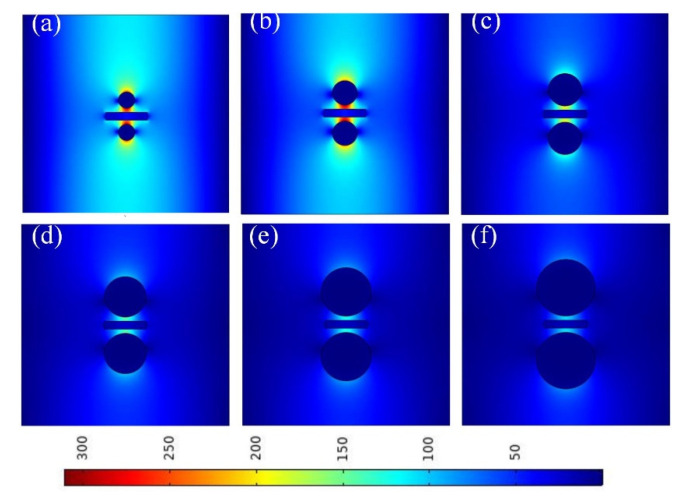
Electric field distributions at different values of *r_Ag_*: (**a**) *r_Ag_* = 20 nm; (**b**) *r_Ag_* = 30 nm; (**c**) *r_Ag_* = 40 nm; (**d**) *r_Ag_* = 50 nm; (**e**) *r_A_*_g_ = 60 nm; and (**f**) *r_Ag_* = 70 nm.

**Figure 8 micromachines-13-01009-f008:**
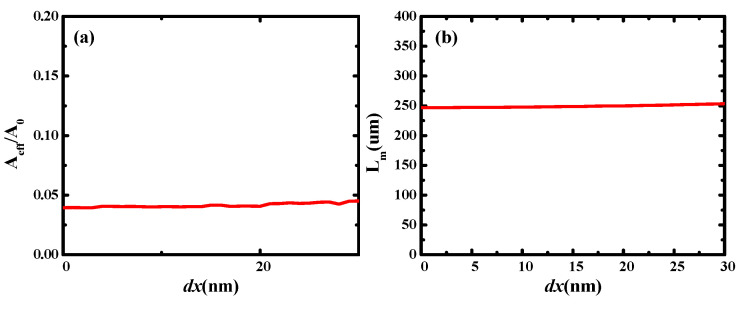
Dependence of waveguide properties on central symmetric error (*dx*): (**a**) normalized mode area (*A_eff_*/*A*_0_); (**b**) propagation length (*L_m_*); (**c**) quality factor of waveguide (*FOM*); and (**d**) modal effective index (*n_eff_*).

**Figure 9 micromachines-13-01009-f009:**
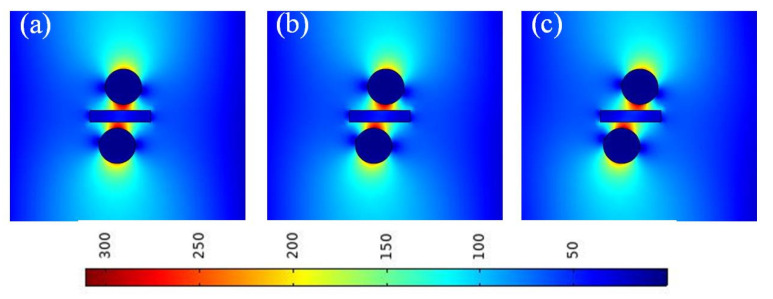
Electric field distributions at different values for *dx*: (**a**) *dx* = 10 nm; (**b**) *dx* = 20 nm; and (**c**) *dx* = 30 nm.

## Data Availability

Not applicable.

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
