# Peer review of "Hybrid Nanowire–Rectangular Plasmonic Waveguide for Subwavelength Confinement at 1550 Nm"

_micromachines, 2022, doi:10.3390/mi13071009_

Round 1

Reviewer 1 Report

Interesting and good written paper.

I suggest to reduce the number of pictures in Fig. 5 to three - for Si thicknesses 5, 15 and 30 nm, the changes are so small, that for illustration it is enough. Maybe the same could be done with the Fig.9.

I am missing the figure of the final structure of proposed waveguide after fabrication, i.e. with the mentioned dimensions 500 x 500 nm.

Author Response

Reviewer 1:

Point #1: I suggest to reduce the number of pictures in Fig. 5 to three - for Si thicknesses 5, 15 and 30 nm, the changes are so small, that for illustration it is enough. Maybe the same could be done with the Fig.9:

Respond:

Thank you very much. We have reduced the number of pictures in Fig. 5 and Fig.9 to three.

Point #2: I am missing the figure of the final structure of proposed waveguide after fabrication, i.e. with the mentioned dimensions 500 x 500 nm.

       Respond:

Thank you very much. In the manuscript, the design and simulation research of the hybrid waveguide is mainly carried out, and it has not been manufactured yet. Further efforts will be made to realize the manufacturing of the waveguide.

Reviewer 2 Report

Yindi et al. report a symmetric hybrid waveguide consisting of two silver (Ag) nanowires and a rectangular silicon (Si) waveguide, where long transmission distance of 270 µm and effective mode area of 0.01 were reported. Although the paper was presented well, the following issues need to be addressed.

(1)    Plasmonic waveguides have been widely investigated at different wave bands based on different materials. However, references in Introduction were less relative to this fields as well as did not reflect the recent development. Such as nature photonics 2.8 (2008): 496-500; Optics express, 2009, 17(19): 16646-16653; Optical Materials, 2022, 128: 112436; Optics Letters, 2021, 46(3): 472-475; Journal of Materials Chemistry C 8.20 (2020): 6832-6838; Nanomaterials, 2021, 11(1): 210, Nanomaterials 10.2 (2020): 229.…

(2)    Aeff/A0 [1] is defined as normalized effective mode area, Aeff [m^2]is the effective mode area; These two variables have different units. For instance, “small effective mode area (Aeff/A0: 0.01–0.05)” should be “small normalized effective mode area (Aeff/A0: 0.01–0.05)”; Also in Abstract, normalized effective mode area of 0.01…

(3)    In Figure 4, we see the hybrid waveguide were only slightly affected by the Si thickness, which is not the same as shown in hybrid waveguides based on graphene (Nanomaterials, 2021, 11(5): 1281).

(4)    Figure 6, Dependence of waveguide characteristics on radius of Ag nanowires (rAg), the x direction should be rAg.

(5)    The width of the Si waveguide is not provided, which is very important. What about the results shown in Figure 6 if the width of the Si waveguide is far larger than 2* rAg?

(6)    The author should tone down their claim (ultra-deep subwavelenth) in the title, since Aeff/A0 is only 0.01.

(7)    The styles of references should be greatly improved.

(8)    Variables in text and figures should be italic.

Author Response

Reviewer 2:

Point #1: Plasmonic waveguides have been widely investigated at different wave bands based on different materials. However, references in Introduction were less relative to this fields as well as did not reflect the recent development. Such as nature photonics 2.8 (2008): 496-500; Optics express, 2009, 17(19): 16646-16653; Optical Materials, 2022, 128: 112436; Optics Letters, 2021, 46(3): 472-475; Journal of Materials Chemistry C 8.20 (2020): 6832-6838; Nanomaterials, 2021, 11(1): 210, Nanomaterials 10.2 (2020): 229.…

Respond:

       Thank you very much! We have added the relevant literatures in the manuscript.

[4]     R. F. Oulton.;V. J. Sorger.;D. A. Genov.;D. F. P. Pile.;X. Zhang.A hybrid plasmonic waveguide for subwavelength confinement and long-range propagation. Nature Photonics. 2008,vol. 2, pp. 496-500.

[5]     D. Dai.;S. He.A silicon-based hybrid plasmonic waveguide with a metal cap for a nano-scale light confinement. Optics Express. 2009,vol. 17, pp. 16646-53.

[11]   X. He.;F. Liu.;F. Lin.;W. Shi.Tunable 3D Dirac-semimetals supported mid-IR hybrid plasmonic waveguides. Optics letters. vol. 46, pp. 472-475.

[12]   D. Teng.;K. Wang.;Q. Huan.;W. Chen.;Z. Li.High-performance light transmission based on graphene plasmonic waveguides. Journal of Materials Chemistry C. 2020,vol. 8,

[13]   D. Teng.;K. Wang.Theoretical Analysis of Terahertz Dielectric–Loaded Graphene Waveguide. Nanomaterials. 2021,vol. 11, p. 210.

[17]  T. A. Da.;A. Zw.;B. Qh.;A. Hw.;W. C. Kai.A low loss platform for subwavelength terahertz graphene plasmon propagation. Optical Materials. vol. 128,

[23]   D. Teng.;K. Wang.;Z. Li.Graphene-Coated Nanowire Waveguides and Their Applications. Nanomaterials. 2020,vol. 10.

Point #2: Aeff/A0 [1] is defined as normalized effective mode area, Aeff [m^2]is the effective mode area; These two variables have different units. For instance, “small effective mode area (Aeff/A0: 0.01–0.05)” should be “small normalized effective mode area (Aeff/A0: 0.01–0.05)”; Also in Abstract, normalized effective mode area of 0.01…

Respond:

Thank you very much! You're right. Aeff/A0 and Aeff are two variables which have different units. We have corrected these points in the manuscript.

Point #3: In Figure 4, we see the hybrid waveguide were only slightly affected by the Si thickness, which is not the same as shown in hybrid waveguides based on graphene (Nanomaterials, 2021, 11(5): 1281):

Respond:

This is a very good question. Thank you very much. In Figure 4, the hybrid waveguide is only slightly affected by the Si thickness, but there is a trend of change. The main reason is that the overall size of our device is small, so we only simulate the waveguide characteristics on TSi in the range of 1-30 nm. The comparison figure of transmission distance between the two literatures is shown below:

Point #4:  Figure 6, Dependence of waveguide characteristics on radius of Ag nanowires (rAg), the x direction should be rAg.

Respond:

Thank you very much. This is our mistake and we have corrected it in the manuscript.

Point #5: The width of the Si waveguide is not provided, which is very important. What about the results shown in Figure 6 if the width of the Si waveguide is far larger than 2* rAg?

Respond:

Thank you very much. We have added instructions for the width of the Si waveguide in manuscript. When the width of the Si waveguide is far larger than 2* rAg, due to the increase of the size of Si waveguide, the distance between the two sides of silicon waveguide and silver nanowires becomes larger, resulting in no strong coupling between the two modes. The waveguide electric field distribution with different Si waveguide widths (100, 200, 300, 400, 500 nm) are simulated as shown in the figures below. In this simulation, rAg=30 nm. We can see that when the width of the Si waveguide is far larger than 2* rAg, the electric field is not concentrated in the coupling region, it's scattered in the waveguide.

Point #6: The author should tone down their claim (ultra-deep subwavelenth) in the title, since Aeff/A0 is only 0.01.

Respond:

Thank you very much. We have changed the title of the article to ”Hybrid nanowire-rectangular plasmonic waveguide for subwavelength confinement at 1550 incident wave-length”.

Point #7: The styles of references should be greatly improved.

Respond:

Thank you very much. We have checked all references and corrected the styles of references.

Point #8: Variables in text and figures should be italic.

Respond:

You are right. Variables in text and figures should be italic. We have corrected it.

Round 2

Reviewer 2 Report

The authors  responded well about the Reviewer's concerns.